# High-Sensitivity Cardiac Troponin [hs-cTn] as a Valuable Biomarker for Pulmonary Hypertension Risk Stratification: A Contemporary Review of the Literature

**DOI:** 10.3390/healthcare12202037

**Published:** 2024-10-14

**Authors:** Vijay Durga Pradeep Ganipineni, Sahas Reddy Jitta, Mohan Chandra Vinay Bharadwaj Gudiwada, Jaswanth Rao Jasti, Chaitra Janga, Bhavyasri Merugu, Revanth Reddy Bandaru, Srikanth Puli, Vikramaditya Samala Venkata, Advait Vasavada, Rupak Desai

**Affiliations:** 1Department of Medicine, Andhra Medical College, Visakhapatnam 530002, India; 2Department of Internal Medicine, Mercy Hospital, Saint Louis, MO 63141, USA; 3Department of Internal Medicine, University of Nebraska Medical Center, Chicago, IL 60657, USA; 4Department of Internal Medicine, University of South Dakota, Sioux Falls, SD 57108, USA; 5Department of Internal Medicine, Jefferson Abington Hospital, Abington, PA 19001, USA; 6Department of Medicine, MediCiti Institute of Medical Science, Hyderabad 501401, India; 7Department of Internal Medicine, East Carolina University, Greenville, NC 27834, USA; 8Department of Hospital Medicine, Cheshire Medical Center, Keene, NH 03431, USA; 9Department of Family Medicine, University of Nebraska Medical Center, Omaha, NE 68198, USA; advait2163@gmail.com; 10Independent Researcher, Atlanta, GA 30079, USA; drrupadesai@gmail.com

**Keywords:** pulmonary hypertension/pulmonary artery hypertension, high-sensitivity troponin, high-sensitivity cardiac troponin, mortality/outcomes, heart failure

## Abstract

**Background:** Pulmonary hypertension (PH) can lead to cardiac failure, thereby significantly affecting life expectancy and quality of life. Due to inadequate disease surveillance and risk assessment, clinical challenges persist despite advances in diagnosis and treatment. We aimed to review the potential of high-sensitivity cardiac troponin (hs-cTn) as a biomarker for predicting outcomes in PH patients. **Methods:** A thorough examination of the PubMed and Google Scholar databases was conducted through March 2023. Studies involving adult PH patients and hsTn as a prognostic indicator of outcomes such as mortality, hospitalization, and disease progression were included, after screening their titles and abstracts. Two independent evaluators extracted data, with the quality assessed using the JBI critical appraisal tool. **Results:** This review uncovered eight studies that examined the prognostic value of hs-cTn in PH patients. Higher hs-cTn levels were associated with increased mortality and hospitalization rates, according to the studies. The severity of PH, cardiac dysfunction, right ventricular function, and systolic dysfunction were associated with hs-cTn. Multiple studies have demonstrated that hsTn has the potential to identify high-risk PH patients who could benefit from targeted therapies and increased clinical monitoring. **Conclusions:** This review suggests that hsTn may be a biomarker for PH risk stratification and prognosis. Across PH subtypes, elevated hsTn levels predict poor outcomes. However, large-scale prospective studies are needed to confirm hs-cTn’s function in diagnosing pulmonary hypertension and determine its potential value in treatment.

## 1. Introduction

Pulmonary hypertension (PH) is a debilitating and life-threatening condition characterized by elevated blood pressure in the pulmonary arteries, which can cause heart failure, ultimately leading to mortality [1]. Since PH is a progressive condition, a patient’s life expectancy and quality of life may be markedly reduced [2]. With few alternatives for illness monitoring and risk assessment, PH continues to pose a substantial clinical challenge, despite advances in diagnosis and therapy [3]. Once pulmonary hypertension develops, it is very difficult to treat and presently impossible to cure. The early identification of patients at high risk of adverse outcomes is crucial for the optimal management of PH [3]. Patients with acute heart failure have a poor prognosis when certain cardiorenal biomarkers, such as troponins, B-type natriuretic peptides (BNPs), and neutrophil gelatinase-associated lipocalins, are elevated [4]. It is unclear whether these markers are associated with adverse events in chronic right ventricular dysfunction due to PH, or whether their measurement may improve risk assessments in the outpatient setting [5].

High-sensitivity troponin (hs-cTn) is a biomarker that has promising early reports as a prognostic tool for various cardiovascular diseases [6]. It indicates the severity of cardiac injury and is secreted as a result of myocardial damage and several other mechanisms [6]. See Figure 1 for its potential mechanisms. The potential predictive utility of hs-cTn for PH is currently gaining attention, since it may shed light on the biology of the condition and help identify patients who are at risk for disease progression and adverse outcomes [7]. Previous studies have explored the role of hs-cTn in PH, and the results have been mixed. Some studies have shown a significant association between elevated hs-cTn levels and adverse outcomes, such as mortality, hospitalization, and disease progression, while others have found no significant association. The reasons for these discrepancies are not entirely clear, and further research is needed. The objective of this review is to summarize the most recent data regarding hs-cTn’s ability to predict outcomes in patients with PH.

## 2. Materials and Methods

In this review, we thoroughly screened and reviewed titles and abstracts for eligibility from two major electronic databases, PubMed and Google Scholar. We used a combination of relevant keywords, including “high-sensitivity troponin”, “pulmonary hypertension”, “pulmonary arterial hypertension”, “prognosis”, “mortality”, and “outcomes”. The scope of the search was from the inception of the databases up to March 2023. In addition to an electronic search, we also manually screened the reference lists of the identified studies and relevant reviews to identify any additional relevant studies. The search was limited to studies published in English.

After conducting the initial search, two independent reviewers screened the titles and abstracts to identify potentially relevant studies. Reviewer disagreements were resolved through discussion and consensus. Studies that met the following criteria were included: (1) observational studies (cohort, case–control, or cross-sectional) or randomized controlled trials; (2) studies investigating the prognostic role of hs-cTn in patients with PH; (3) studies reporting relevant clinical outcomes, such as mortality, hospitalization, or disease progression; (4) studies involving adult human subjects; and (5) studies with sample sizes > 20. See Figure 2 below depicting the search for studies using the PRISMA chart [8,9].

Two independent evaluators extracted the data using a standardized data extraction form. Each study’s design, patient characteristics, high-sensitivity troponin assay results, cutoff values, duration of follow-up, outcomes measured, and main results were extracted. Disagreements among reviewers were resolved through discussion and consensus. The quality of the included studies was evaluated with the cohort/observational JBI quality appraisal tool [10]. The results were subjected to a narrative synthesis, and the findings were summarized and presented in a descriptive and tabular format.

## 3. Results

### Review of Literature/Evidence

A total of eight studies were found to be eligible for qualitative synthesis. Table 1 provides a summary of the studies’ characteristics.

Roy et al. [4] executed a cohort investigation involving 108 subjects attending the National Pulmonary Hypertension Unit in Dublin, Ireland, from 2007 to 2009, with the aim of ascertaining the prognosticators of mortality and hospitalization [4]. Over a median observation duration of 4.1 years, 50 patients (46.3%) experienced either death or hospitalization. The investigation established that a decrementing six-minute walk test (6MWT; hazard ratio [HR] 12.8; *p* < 0.001), BNP (HR 6.68; *p* < 0.001), and hsTnT (HR 5.48; *p* < 0.001) were independent prognosticators of mortality. Adjusted hazard analyses retained significance upon incorporating hsTnT into a model with BNP and the 6MWT (HR 9.26, 95% CI 3.61–23.79), as did the predictive capability of the model for death and rehospitalization (area under the receiver operating characteristic curve 0.81, 95% CI 0.73–0.90). These outcomes suggest that hsTnT can discern a pulmonary hypertension subgroup with a bleaker prognosis and might be employed in a risk prediction model to pinpoint patients at elevated risk, who may require the escalation of targeted pulmonary vasodilator therapies and closer clinical monitoring.

The study by Vélez-Martínez et al. [17] was conducted when hsTroponin assays were somewhat new, and it revealed that the highly sensitive cTnI assay detected troponin I in 95% of patients, which is significantly higher than traditional assays. This revealed that even low levels of myocardial injury might be more common in pulmonary hypertension (PH) patients than previously thought. Elevated cTnI levels, particularly in the highest quartile, were strongly associated with a much higher risk of death (adjusted hazard ratio of 5.3), suggesting that cTnI could be a more potent predictor of mortality than other traditional markers in PH. Furthermore, higher cTnI levels correlated with more severe hemodynamic and cardiac abnormalities, as seen on cardiac magnetic resonance imaging (MRI), indicating that even slight increases in cTnI could reflect significant underlying cardiac damage in PH patients.

Schuuring et al. [11] aimed to determine whether hsTnT, measured during a routine outpatient visit, correlated with the prognosis of pulmonary arterial hypertension due to congenital heart disease (CHD-PAH) [11]. The study encompassed 31 subjects (mean age 45 ± 12 years) with CHD-PAH referred for advanced medical therapy between January 2005 and March 2007 at the Academic Medical Center in Amsterdam. During a median follow-up of 5.6 years, eight patients succumbed. Elevated hsTnT concentrations were observed in eight patients (26%). In a univariate Cox regression, baseline-elevated hsTnT, NT-pro-BNP, and right ventricular function were determinants of death (*p* < 0.05 for all). Patients with elevated hsTnT concentrations exhibited a significantly higher mortality rate compared to those with normal hsTnT concentrations (62% vs. 13%, *p* = 0.005). These findings suggest that hsTnT is associated with prognosis in CHD-PAH patients and elevated hsTnT concentrations are a predictor of mortality.

Kvisvik and colleagues [12] investigated whether high-sensitivity cardiac troponin T (hs-cTnT) concentrations correlated with prognosis independently of pulmonary hypertension and ventricular dysfunction in patients with stable chronic obstructive pulmonary disease (COPD) [12]. The cohort consisted of 112 patients with GOLD stage I–IV, and hs-cTnT was measured in 98 patients. Over a mean follow-up of 7.8 ± 3.0 years, 49 deaths occurred. The authors discovered that hs-cTnT was detectable in 87% of the measured samples, while the median value was 7 ng/L (interquartile range: 4–10). Elevated levels of hs-cTnT were correlated with right ventricular (RV) TAPSE, RV myocardial performance index, and left ventricular strain. Troponin concentrations were significantly increased in patients with pulmonary hypertension and an increasing GOLD stage. The authors identified a significant association between hs-cTnT and all-cause mortality, which remained significant when adjusting for pulmonary hypertension and indices of systolic dysfunction. When the analyses were performed according to sex, the associations remained significant in men but not in women. This study suggests that elevated hs-cTnT levels are associated with the severity of pulmonary hypertension and cardiac dysfunction in patients with stable COPD and are an independent predictor of all-cause mortality.

Kimura and colleagues [13] endeavored to explore the clinical implications of cardiac troponin in patients with chronic thromboembolic pulmonary hypertension (CTEPH) undergoing balloon pulmonary angioplasty (BPA) treatments. They scrutinized 63 sequential CTEPH patients who received BPA procedures and assessed high-sensitivity troponin T (hsTnT) concentrations pre- and post-BPA. The researchers discovered that initial hsTnT values correlated with older age and were elevated in individuals experiencing increased right atrial pressure, average pulmonary artery pressure, and pulmonary vascular resistance and diminished pulmonary capillary wedge pressure and six-minute walk distance. They observed significant enhancements in mean pulmonary artery pressure and pulmonary vascular resistance, as well as a marked reduction in hsTnT following BPA. Based on the alterations in hsTnT after BPA, patients were categorized into the hsTnT-decrease group (*n* = 34) and hsTnT-increase or stable group (*n* = 29). The hsTnT-decrease group exhibited more considerable reductions in mean pulmonary artery pressure and pulmonary vascular resistance, leading to improved outcomes compared to the hsTnT-increase or stable group. Kriechbaum et al. [16] and Kimura et al. [13] both explored the role of hs-cTnT as a biomarker of myocardial damage in patients with CTEPH undergoing BPA. Both studies are in agreement regarding the utility of hs-cTnT in monitoring the therapeutic impact of BPA, though Kriechbaum et al. expanded this analysis by also considering NT-proBNP, providing a comprehensive picture during treatment.

Heresi and co-authors [14] aimed to examine whether a cardiac troponin I (cTnI) measurement, utilizing a sensitive assay, correlated with disease severity and prognosis in pulmonary arterial hypertension (PAH) patients. In their study, they included 68 PAH patients classified as diagnostic category 1 and measured cTnI by employing a sensitive immunoassay with a lower detection limit of 0.008 ng/mL. Their findings demonstrated that cTnI was detected in 25% of PAH patients. Those with detectable cTnI exhibited more severe functional class symptoms, a reduced six-minute walk distance, increased pericardial effusions, an enlarged right atrial area, and elevated B-type natriuretic peptide and C-reactive protein levels. Moreover, the 36-month transplant-free survival rate was 44% for patients with detectable cTnI compared to 85% for those with undetectable cTnI. Importantly, cTnI was linked to a 4.7-fold higher risk of death due to right ventricular failure or transplant, even after adjusting for known PAH severity parameters. Consequently, the study concluded that employing a sensitive assay to measure cTnI can forecast disease severity and prognosis in PAH patients and identify individuals at increased risk of death related to right ventricular failure or transplant.

Filusch and associates [15] sought to evaluate the novel high-sensitive cardiac troponin T (hsTnT) assay’s usefulness for functional and prognostic assessments in patients diagnosed with PAH. They included 55 PAH patients with an average pulmonary artery pressure of 45 ± 18 mmHg in their study and assessed cTnT levels using both a conventional fourth-generation assay and the new hsTnT assay, which had a lower detection limit of 2 pg/mL. The outcomes revealed that the hsTnT assay was superior to the fourth-generation assay in detecting cTnT levels, with 90.9% of patients having detectable cTnT using the hsTnT assay, compared to just 30.9% with the fourth-generation assay. Concentrations exceeding the 99th percentile were found in 27.3% of patients using the hsTnT assay versus 10.9% using the fourth-generation assay. Patients with elevated hsTnT levels had a worse prognosis, with five of the six patients with cTnT values > 29.5 pg/mL dying during the 12-month follow-up. Additionally, a correlation was identified between hsTnT levels and other indicators of disease severity, such as a reduced six-minute walk distance, more advanced WHO functional class, and right ventricular systolic dysfunction. In the area under the curve (AUC) analysis, hsTnT proved to be as effective as established biomarkers like heart-type fatty-acid-binding protein (hFABP) or N-terminal pro-brain natriuretic protein (NT-proBNP) in predicting mortality. Furthermore, hsTnT demonstrated a better predictive ability for a WHO functional class > II compared to hFABP or NT-proBNP. In conclusion, the authors posited that the hsTnT assay serves as a valuable instrument for evaluating prognosis and disease severity in PAH patients. Increased hsTnT concentrations were linked to adverse outcomes, including death, and other disease severity parameters, such as impaired exercise tolerance and right ventricular dysfunction. Therefore, the hsTnT assay may prove beneficial in managing and assessing PAH patients.

A quality appraisal of all studies has been summarized in Table 2.

## 4. Discussion

With advances in cardiac troponin testing, hs-cTnT and hs-cTnI have become valuable tools in practical medicine. Over time, as low-sensitive troponin testing has been replaced by high-sensitive troponin tests, it has become possible to interpret different levels of measurements, and non-cardiac causes of troponin elevation became of particular interest to the scientific community [18]. Troponin elevation has since been found to be present in chronic kidney disease, pulmonary embolism, sepsis, stroke, COPD, extreme physical exercise, burns, ARDS, rhabdomyolysis, and chemotherapy [19,20]. In addition, structural heart diseases and arterial and pulmonary hypertension are of recent interest [21]. Elevated troponin levels alone do not definitively point to one condition over another, as all these conditions can cause some myocardial damage or elevate troponin levels via a particular mechanism. Thus, it is not recommended for use as a primary diagnostic marker. For example, similar to PH, pulmonary embolism can cause increased right ventricular strain, leading to elevated troponin levels. This makes it hard to distinguish between PH and PE based solely on troponin. Hence, the marker should be used not for primary diagnosis, but only to assess potential risk, taking into account other factors in the patient’s history; co-morbidities; severity, if already diagnosed with pulmonary hypertension; and prognosis [18,19,20].

Clinical studies confirm that hypertension is a major contributor to elevated hs-cTnT and hs-cTnI levels [22]. These biomarkers are highly predictive, helping identify patients with prehypertensive conditions, hypertension, and hypertensive crisis who are at increased risk for short- and long-term cardiovascular events [21]. The increase in hs-cTnT and hs-cTnI in arterial hypertension is attributed to myocardial tissue distension, proteolytic degradation, and increased cardiomyocyte membrane permeability [23]. Additionally, factors such as blood pressure’s impact on troponin elimination, myocardial hypertrophy, and left ventricle wall stress contribute to these elevations. Although all of these hypotheses are being explored, pulmonary hypertension may also have a similar consequence. In clinical practice, the ability to detect minor and subclinical myocardial damage through highly sensitive and ultrasensitive assays for cTnT and cTnI is crucial. Taking a step further, the early identification of patients at risk during prehypertensive stages, as well as those with pulmonary hypertension, may enable timely intervention with targeted therapeutic and preventive measures to reduce the risk of severe complications [24]. The use of hs-Troponin in cases of pulmonary hypertension can lead to the emergence of necessary preventive measures.

Our review of the literature included relevant studies that consistently demonstrated the prognostic value of high-sensitivity cardiac troponin (hs-cTnT) in various populations of pulmonary hypertension (PH) patients. These findings illustrate that hs-cTnT may serve as an important biomarker in assessing the severity of PH and predicting patient outcomes, including mortality and hospitalization.

Roy et al. [4] demonstrated that hsTnT, when added to a model with BNP and 6MWT, independently predicted mortality and rehospitalization in patients attending the National Pulmonary Hypertension Unit in Dublin, Ireland. Schuuring et al. [11] found that elevated hsTnT levels were associated with a significantly higher mortality rate in patients with pulmonary arterial hypertension due to congenital heart disease (CHD-PAH). Kvisvik et al. [12] provided evidence that elevated hs-cTnT levels are associated with the severity of pulmonary hypertension and cardiac dysfunction in patients with stable chronic obstructive pulmonary disease (COPD) and are independent predictors of all-cause mortality. Kimura et al. [13] showed that hsTnT levels improved following balloon pulmonary angioplasty (BPA) in chronic thromboembolic pulmonary hypertension (CTEPH) patients, and that patients with a decrease in hsTnT had better outcomes than those with stable or increased hsTnT levels.

Heresi et al. [14] demonstrated that cTnI levels, measured using a sensitive assay, were associated with disease severity and prognosis in patients with pulmonary arterial hypertension (PAH). Filusch et al. [15] found that a novel hsTnT assay was more effective in detecting cTnT levels in PAH patients, and that elevated hsTnT levels correlated with poor prognosis and other parameters of disease severity.

Taken together, these studies indicate that high-sensitivity cardiac troponin assays, whether measuring cTnT or cTnI, can provide valuable information for the management and evaluation of patients with PH. The studies also suggest that hsTnT may play a role in identifying patients who may benefit from targeted pulmonary vasodilator therapy and closer clinical monitoring. These results suggest that hsTnT may be a useful tool for identifying patients at higher risk for side effects who might benefit from targeted interventions. Our findings are consistent with a meta-analysis conducted by Xu et al. [25]. However, several studies in that meta-analysis used older-generation troponin studies, which were less sensitive. An interesting finding in their meta-analysis was that American populations had increased mortality compared to European populations. Hence, regional differences remain to be explored rigorously. The consistent association between elevated hs-cTn levels and poor prognosis illustrates the potential utility of hs-cTnT as a biomarker for risk stratification and prognostication in PH patients. Furthermore, the independent predictive value of hs-cTn indicates that it may serve as a complementary tool to existing biomarkers, such as BNP and 6MWT, in assessing PH severity and predicting patient outcomes.

The studies in our review vary in terms of the marker used. Some have looked at TnT, while in others TnI has been studied. TnI is degraded and released faster from necrotic cardiac tissue compared to TnT, leading to higher peak concentrations and a faster return to baseline [18,19]. This characteristic is important in identifying acute ischemic events, but it remains to be known how this could play into the risk stratification of pulmonary hypertension. TnT elevation has a higher co-relation to chronic kidney disease compared to TnI [19,20]. This may complicate its interpretation in a subset of patients with PH.

It is important to acknowledge the limitations of the studies included in this review, such as their relatively small sample sizes and single-center designs. The sample sizes in our study ranged from 30 to 240 s, thereby limiting generalizability. Larger, multicenter studies are thus needed to validate the prognostic value of hs-cTn in PH patients and to establish optimal cut-off values and measurement intervals for clinical application. The development of such models would be invaluable in guiding personalized treatment strategies and improving patient outcomes in PH. Also, the studies had different inclusion criteria, outcome measures, and follow-up periods, which make direct comparisons challenging, and they did not investigate the role of hsTnT in the diagnosis of pulmonary hypertension and its diagnostic accuracy remains unclear. As one reviewer suggested, our study lacks data analysis and sensitivity testing of the included studies, which were not performed due to the heterogenous nature of the studies, whose methods varied widely according to the authors. Finally, this study highlights the need for future research to explore the potential of combining hs-cTn with other biomarkers and clinical parameters to develop more accurate and comprehensive risk prediction models for PH patients.

## 5. Conclusions

The findings of these eight studies suggest that hsTn is a promising biomarker for risk classification and prognosis in individuals with pulmonary hypertension. Patients with pulmonary hypertension had higher than normal levels of hsTn and this was related to unfavorable outcomes, such as death and hospitalization. Additionally, the severity of PH, cardiac dysfunction, RV function, and indicators of systolic dysfunction were all linked to hsTn levels. Hence, hsTn may be helpful for pulmonary hypertension patients’ risk classification and prognosis. Larger prospective investigations are necessary to verify these results and determine the function of hsTn and how TnT and TnI, albeit different in character, can be used in determining the risk assessment, severity, and prognosis of pulmonary hypertension. Future studies should also look into the possible benefits of employing hsTn to identify patients who could benefit from certain treatments.

## Figures and Tables

**Figure 1 healthcare-12-02037-f001:**
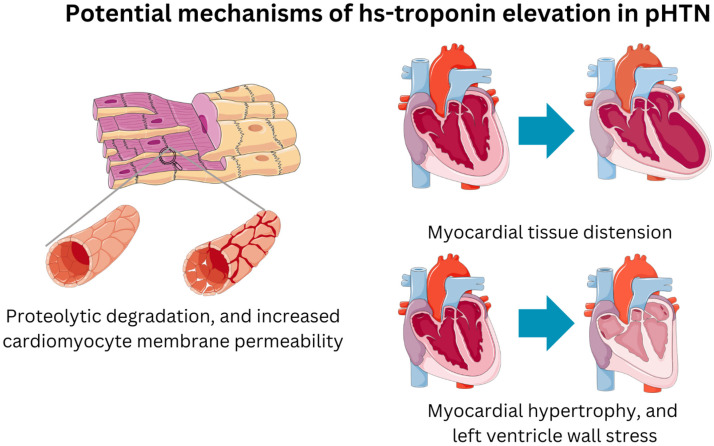
Mechanisms of hs-Troponin elevation in pulmonary hypertension.

**Figure 2 healthcare-12-02037-f002:**
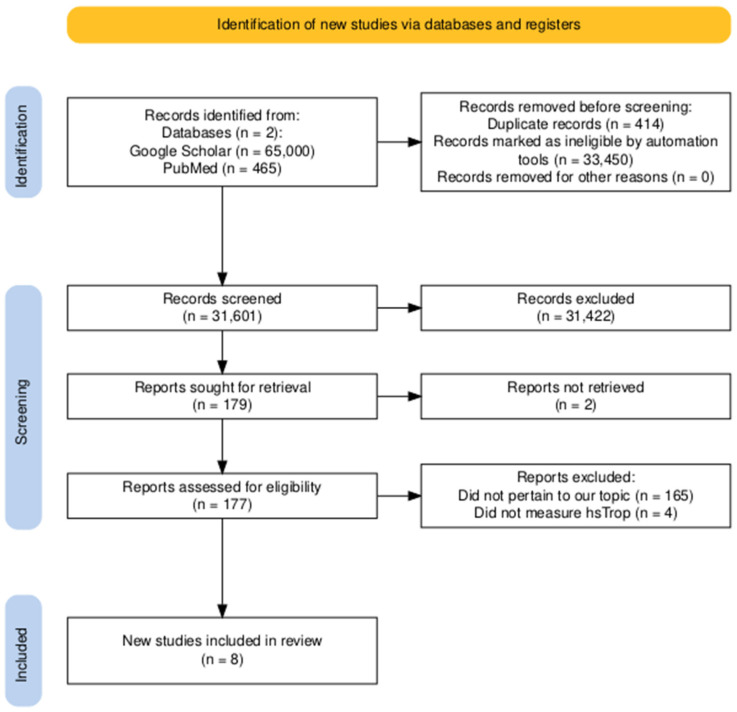
PRISMA chart.

**Table 1 healthcare-12-02037-t001:** Study population and study findings.

Study ID	Type of Study	Study Population	Methodology	Study Findings
Roy et al. [4]	Prospective investigation	108 subjects attending the National Pulmonary Hypertension Unit in Dublin, Ireland (2007–2009). Connective tissue disease-associated pulmonary hypertension was the most common etiology, present in 33 patients.	Primary outcomes were all-cause mortality and combined clinical endpoint of rehospitalization due to pulmonary hypertension.	hsTnT, BNP, and decrementing 6MWT were independent prognosticators of mortality; hsTnT can discern a PH subgroup with poorer prognosis and might be employed in a risk prediction model.
Schuuring et al. [11]	Prospective investigation	31 subjects with CHD-PAH referred for advanced medical therapy at the Academic Medical Center in Amsterdam. All patients were started on bosentan monotherapy.	The primary outcome was all-cause mortality. Patients with severe renal impairment (eGFR < 30 mL/min) were excluded.	Elevated hsTnT concentrations were observed in 26% of patients; elevated hsTnT concentrations were a predictor of mortality.
Kvisvik et al. [12]	Prospective investigation	112 patients with GOLD stage I–IV COPD were selected for this study.	Hs-cTnT was measured in 98 patients and associations with all-cause mortality were assessed by Cox regression and Kaplan–Meier analysis.	Elevated hs-cTnT levels were associated with the severity of pulmonary hypertension and cardiac dysfunction in patients with stable COPD and were an independent predictor of all-cause mortality.
Kimura et al. [13]	Prospective investigation	63 consecutive chronic thromboembolic pulmonary hypertension (CTEPH) patients who underwent BPA procedures.	Measured hsTnT levels before and after BPA.	Significant improvements in mean pulmonary artery pressure and pulmonary vascular resistance, and significant decrease in hsTnT after BPA; hsTnT-decrease group showed better outcomes compared to hsTnT-increase or stable group.
Heresi et al. [14]	Prospective investigation	68 patients with PAH diagnostic category 1.	cTnI was measured using a sensitive immunoassay with a lower limit of detection of 0.008 ng/mL. Acute coronary syndromes and advanced renal disease, defined as a serum creatinine > 2 mg/dL^−1^, were excluded.	Detectable cTnI in 25% of PAH patients; patients with detectable cTnI had more advanced functional class symptoms, shorter six-minute walk distance, and higher mortality risk; cTnI was associated with a 4.7-fold increased risk of death related to right ventricular failure or transplant.
Filusch et al. [15]	Prospective investigation	55 PAH patients with a mean pulmonary artery pressure of 45 ± 18 mmHg.	cTnT levels assessed using both a conventional fourth-generation assay and the novel hsTnT assay with a lower limit of detection at 2 pg/mL.	hsTnT assay was more effective in detecting cTnT levels than the fourth-generation assay; patients with higher hsTnT levels had a poorer prognosis; hsTnT was as effective as established biomarkers (hFABP or NT-proBNP) in predicting death and demonstrated a better predictive ability for WHO functional classes > II; hsTnT assay may be valuable in evaluating prognosis and disease severity in PAH patients.
Kriechbaum et al. [16]	Post hoc analysis	51 patients with chronic thromboembolic pulmonary hypertension (CTEPH).	hs-cTnT and NT-proBNP levels before each balloon pulmonary angioplasty (BPA) and at a 6-month follow-up.	Steady decrease in hs-cTnT and NT-proBNP levels, with significant reductions in mean pulmonary arterial pressure (meanPAP) and pulmonary vascular resistance (PVR).
Vélez-Martínez et al. [17]	Prognostic observational study	255 patients diagnosed with pulmonary hypertension.	hsTroponin levels, along with demographic data, hemodynamic assessments, cardiac MRI, and B-type natriuretic peptide levels, with survival analyzed using Kaplan–Meier and Cox regression methods.	With a median follow-up of 3.5 years, higher cTnI levels were associated with worse hemodynamics, structural cardiac abnormalities, and increased mortality, providing prognostic information for patients with PH.

**Table 2 healthcare-12-02037-t002:** Quality appraisal using JBI tool for observational/cohort studies.

Studies/Quality Metrics	Roy et al. [4]	Schuuring et al. [11]	Kvisvik et al. [12]	Kimura et al. [13]	Heresi et al. [14]	Filusch et al. [15]	Kriechbaum et al. [16]	Vélez-Martínez et al. [17]
Comparability	Yes	Yes	Unclear	Unclear	Yes	Yes	Yes	Yes
Matching	Yes	No	No	No	Yes	No	Yes	Yes
Standard criteria for identification	Yes	Yes	Yes	Yes	Yes	Yes	Yes	Yes
Reliable and standard tools of measurement	Unclear	Yes	Yes	Yes	Yes	Yes	Yes	Unclear
Consistency in measurement	Yes	Yes	Unclear	Unclear	Yes	Yes	Yes	Yes
Confounding Identified	No	Unclear	Unclear	Unclear	No	Unclear	No	Unclear
Strategies to address confounding	Yes	Unclear	No	No	Yes	Yes	Yes	Yes
Standard assessment method	Yes	Yes	Yes	Yes	Yes	Yes	Yes	Yes
Appropriate duration	No	Yes	Yes	Yes	Unclear	Yes	Unclear	Yes
Appropriate analysis	Yes	Yes	Unclear	Unclear	Yes	Yes	Yes	Yes

## Data Availability

Available on request.

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
