# Peer review of "High-Sensitivity Cardiac Troponin [hs-cTn] as a Valuable Biomarker for Pulmonary Hypertension Risk Stratification: A Contemporary Review of the Literature"

_healthcare, 2024, doi:10.3390/healthcare12202037_

Round 1

Reviewer 1 Report

Comments and Suggestions for Authors

Comments and Suggestions for Authors (will be shown to authors)

In the manuscript titled “High-Sensitivity Cardiac Troponin [hs-cTn] as a Valuable Biomarker for Pulmonary Hypertension Risk Stratification: A Contemporary Review of Literature,” the authors review the existing literature on the use of hs-cTn as a potential marker for pulmonary hypertension using various previously documented cases.

The authors have shown a thorough investigation of the literature and have crafted a strong review manuscript.

Major Comments and Suggestions

A significant aspect the authors should address is the biological and clinical differences between TnT and TnI. This would help direct future research towards a specific marker rather than a group which would further strengthen the utility of this manuscript.

Author Response

Comment 1: A significant aspect the authors should address is the biological and clinical differences between TnT and TnI. This would help direct future research towards a specific marker rather than a group which would further strengthen the utility of this manuscript.

Response 1: Thank you for pointing this out. We have included studies based on both TnT and TnI, but now have briefly described how they are different and it remains to be known in what situations they can be used in pulmonary hypertension. We have revised the conclusion to highlight this fact as well. 

Thank you 

Reviewer 2 Report

Comments and Suggestions for Authors

Elevated troponin is generally associated with increased cardiovascular events in patients with hypertension crisis throughout the world. Which among the following will be a most probable indicator if the diagnosis is not properly carried out? 

Pulmonary hypertension/Pulmonary embolus/Congestive heart failure/ Long-term kidney disease/ Cardiomyopathy

Does it depend upon the food style?

 Have the authors accounted for the nature of the work profile linked with job working hours related to the profession. this is of importance since the blood pressure is directly related to the nature of the geographical range and its influences accompanied with the profile of the various life style. How could these could be correlated  

Author Response

Comment1: 

Elevated troponin is generally associated with increased cardiovascular events in patients with hypertension crisis throughout the world. Which among the following will be a most probable indicator if the diagnosis is not properly carried out? 

Pulmonary hypertension/Pulmonary embolus/Congestive heart failure/ Long-term kidney disease/ Cardiomyopathy

Response 1: Thank you for your comment. I agree that many conditions may affect the findings. Hence, to clarify these points we have revised the discussion to include "]. Elevated troponin levels alone do not definitively point to one condition over another, as all these conditions can cause some myocardial damage or elevate troponin levels via a particular mechanism. Hence it is not recommended to use it as a primary diagnostic marker. For example, similar to PH, pulmonary embolism can cause increased right ventricular strain, leading to elevated troponin levels. This makes it hard to distinguish between PH and PE based solely on troponin. Hence, the marker should be used not for primary diagnosis but only to assess potential risk taking into account other factors in the patient history, co-morbidities, severity if already diagnosed with pulmonary hypertension and prognosis ". Additionally, we have made a few more minor edits to the conclusion and discussion to highlight this point further

Comment 2:  Have the authors accounted for the nature of the work profile linked with job working hours related to the profession. this is of importance since the blood pressure is directly related to the nature of the geographical range and its influences accompanied with the profile of the various life style. How could these could be correlated  

Response 2: Although diet and lifestyle contribute to cardiovascular diseases, exploring a direct link with pulmonary hypertension was out of the scope for our topic. Plus, most of the evidence currently points to the indirect relationship. Food choices and BMI may contribute to the development of hypertension or exacerbate existing cardiovascular disease, they are less directly linked to pulmonary hypertension. Based on current evidence lifestyle could exacerbate associated conditions like left heart disease or chronic lung disease, which may indirectly worsen PH. However, the direct impact of these factors on troponin levels in PH patients would likely be more related to overall cardiovascular health rather than lifestyle alone. Hence, even though it is a very important idea, we couldn't include it in our manuscript 

Thank you

Reviewer 3 Report

Comments and Suggestions for Authors

The proposed project sought to review the potential of high-sensitivity cardiac troponin (hs-cTn) as a biomarker for predicting outcomes in pulmonary hypertension patients. The current proposal is interesting and well-written. Therefore, I recommend that the current study be published after major revisions as follows:

1-   Could the authors provide the PRISMA flow diagram?

2-   Please highlight sensitivity testing for all research included in this review.

3-    Please add a diagrammatic figure to propose the possible mechanistic pathway of high-sensitivity cardiac troponin on pulmonary hypertension patients

4-   Please check the manuscript for typos and error 

Author Response

Thank you for your generous comments, we looked at each of your point individually and responded to them. 

Comment 1-   Could the authors provide the PRISMA flow diagram?

We have included the PRISMA chart in our article. Citations to PRISMA and software used cited in references

2-   Please highlight sensitivity testing for all research included in this review.

Due to the narrative nature of our text and variability in individual studies, we did not perform any meta-analysis. We have included this in the limitations of our article

3-    Please add a diagrammatic figure to propose the possible mechanistic pathway of high-sensitivity cardiac troponin on pulmonary hypertension patients

We illustrated the mechanisms of troponin elevation in pulmonary hypertension in the figure and provided acknowledgment to the source of images used to create the figure. 

4-   Please check the manuscript for typos and error 

We checked for typos and corrected them as needed

Thank you again for taking a critical look at our manuscript

Round 2

Reviewer 3 Report

Comments and Suggestions for Authors

The authors succesfully handled all comments